# *Pterodon emarginatus* Seed Preparations: Antiradical Activity, Chemical Characterization, and In Silico ADMET Parameters of β-caryophyllene and Farnesol

**DOI:** 10.3390/molecules28227494

**Published:** 2023-11-09

**Authors:** Guglielmina Froldi, Francesco Benetti, Andrea Mondin, Marco Roverso, Elisa Pangrazzi, Francine Medjiofack Djeujo, Paolo Pastore

**Affiliations:** 1Department of Pharmaceutical and Pharmacological Sciences, University of Padova, 35131 Padova, Italy; francesco.benetti90@gmail.com (F.B.); elisapangrazzi99@gmail.com (E.P.); francine.medjiofackdjeujo@phd.unipd.it (F.M.D.); 2Department of Chemical Sciences, University of Padova, 35131 Padova, Italy; mondin.andrea@gmail.com (A.M.); marco.roverso@unipd.it (M.R.); paolo.pastore@unipd.it (P.P.)

**Keywords:** phenols, terpenes, medicinal plants, ADMETlab web tool, essential oil, traditional medicine, gas chromatography, antioxidants, DPPH assay, ORAC assay, ADMET

## Abstract

The study of medicinal plants and their active compounds is relevant to maintaining knowledge of traditional medicine and to the development of new drugs of natural origin with lower environmental impact. From the seeds of the Brazilian plant *Pterodon emarginatus,* six different preparations were obtained: essential oil (EO), ethanol extract (EthE) prepared using the traditional method, and four extracts using solvents at different polarities, such as n-hexane, chloroform, ethyl acetate, and methanol (HexE, ChlE, EtAE, and MetE). Chemical characterization was carried out with gas chromatography, allowing the identification of several terpenoids as characteristic components. The two sesquiterpenes β-caryophyllene and farnesol were identified in all preparations of *Pterodon emarginatus,* and their amounts were also evaluated. Furthermore, the total flavonoid and phenolic contents of the extracts were assessed. Successively, the antiradical activity with DPPH and ORAC assays and the influence on cell proliferation by the MTT test on the human colorectal adenocarcinoma (HT-29) cell line of the preparations and the two compounds were evaluated. Lastly, an in silico study of adsorption, distribution, metabolism, excretion, and toxicity (ADMET) showed that β-caryophyllene and farnesol could be suitable candidates for development as drugs. The set of data obtained highlights the potential medicinal use of *Pterodon emarginatus* seeds and supports further studies of both plant preparations and isolated compounds, β-caryophyllene and farnesol, for their potential use in disease with free radical involvement as age-related chronic disorders.

## 1. Introduction

Medicinal plants are still used in several countries in traditional medicine and also represent an inexhaustible source of potential new drugs. In addition, they are included in the composition of many widely marketed dietary supplements [1]. In traditional Brazilian medicine, the seeds of *Pterodon emarginatus* Vogel (synonyms *Pterodon pubescens* Benth. and *Pterodon polygaliflorus* Benth., Fabaceae family), in the forms of tincture, infusion, or dry extracts, have been used for centuries in a wide range of diseases, especially as systemic analgesic and anti-inflammatory treatments [2]. *Pterodon emarginatus*, also known by the popular names “sucupira-branca” and “faveira”, is a native tree plant that grows in isolated specimens, up to 10 m high, in the middle region of Brazil [2,3,4]. Inside each fruit is one seed, surrounded by an alveolar structure that contains an amber-yellow resin [4,5]. The most common preparations of *Pterodon emarginatus* are made by macerating the seeds in ethyl alcohol (or high-grade alcoholic beverages) for 24–48 h in a ratio of 1:5; then, after filtration, 40 mL per day are administered orally for 7 days to treat mainly inflammatory-related diseases [4,6,7,8].

Previous studies have considered the chemical characterization of extracts obtained from the seeds of *Pterodon emarginatus,* detecting the typical presence of terpenes and isoflavones [3,9]. In detail, in the essential oil of seeds, the presence of β-caryophyllene (35.9%), β-elemene (15.3%), germacrene D (9.8%), spathulenol (5.9%), α-humulene (6.8%), and bicyclogermacrene (5.5%) was reported [3,6]. Terpene compounds have a variety of biological functions, but their potential therapeutic use has not been extensively studied [10,11]. Therefore, during this investigation, a total of six extracts were obtained from the seeds of *Pterodon emarginatus*, such as the essential oil (EO), the ethanol extract used in traditional medicine (EthE), and four extracts using solvents of different polarities, such as n-hexane, chloroform, ethyl acetate, and methanol (HexE, ChlE, EtAE, and MetE). Furthermore, the activity of two components of the seeds, such as β-caryophyllene and farnesol (Figure 1), was evaluated and compared with that of *Pterodon emarginatus* extracts. β-Caryophyllene is a bicycle sesquiterpene that has been identified in several species of plants [12,13]. A notable peculiarity of this molecule is the butane ring, a very rare structure in nature (Figure 1). Farnesol is an acyclic sesquiterpene alcohol that is widely identified in the essential oils of several plants. It also acts as a precursor to sterol biosynthesis [14,15]. Both β-caryophyllene and farnesol are often used in food supplements as flavoring agents and antimicrobial agents [10,16]. Additionally, they are known for their pharmacological potential, as demonstrated in various in vitro and in vivo studies [11,17,18]. As an example, they can modulate NF-kB signaling, which plays an important role in the pathogenesis of inflammatory diseases and cancer [19,20]. It should be noted that Gertsch et al. discovered that β-caryophyllene is naturally present in *Cannabis sativa* L. and has specific agonist activity on the CB2 receptor [21].

This research focused on the study of six selected extracts obtained from the seeds of *Pterodon emarginatus,* detecting their chemical characterization with gas chromatography analysis and antiradical activity using two different antiradical tests, known as DPPH and ORAC assays, also in comparison with β-caryophyllene and farnesol. Successively, plant-derived preparations and the two phytoconstituents were evaluated in the concentration range of 0.01 µg/mL to 50 µg/mL on the viability of the human colorectal adenocarcinoma HT-29 cell line to study their potential cytotoxicity. Additionally, the pharmacokinetic parameters of absorption, distribution, metabolism, excretion, and toxicity (ADMET) of β-caryophyllene and farnesol were evaluated in silico.

## 2. Results and Discussion

### 2.1. Essential Oil and Extracts of Pterodon emarginatus Seeds

In the first phase of this study, several extracts from *Pterodon emarginatus* seeds were prepared to evaluate the types of phytoconstituents present in them and their activity in vitro. Essential oil was obtained by steam distillation, while the extract used in traditional medicine was obtained by macerating the seeds in ethanol for 24 h. Furthermore, an extraction sequence was performed using solvents of different polarities. In this way, by steam distillation, a light white essential oil (EO) was obtained from the seeds with a dry weight yield of 1.5% *w*/*w*, while a higher yield of 40.9% was achieved using traditional procedures with ethanol (EthE). Furthermore, by sequential extraction of solvents with different polarities, such as n-hexane, chloroform, ethyl acetate, and methanol, four extracts were obtained with yields of 16.9%, 16.7%, 1.4%, and 2.0%, respectively (residual plant matrix 63.0%).

### 2.2. Phytochemicals of Pterodon emarginatus Extracts Detected by Gas Chromatography

Qualitative and quantitative analyses were performed using gas chromatography to determine the types of compounds present in the extracts and, furthermore, the amount of β-caryophyllene and farnesol in each one. Figure 2 shows a gas chromatogram of the EO obtained from the seeds of *Pterodon emarginatus*; the peaks corresponding to β-caryophyllene (1) and farnesol (2) were highlighted, while the peak of the internal standard 1,3,5-triisopropylbenzene was indicated with (IS). Exemplificative chromatograms of the extracts studied in this research are available in the Appendix A. Table 1 reports the compounds detected as a percentage ratio between the peak area of interest and the peak area of IS. Each compound reported was identified by comparing the mass spectrum with that found in the NIST Standard Reference database (Appendix A) [22], the percentage abundance relative to the standard, and the retention time. Then, a quantitative analysis was performed for β-caryophyllene and farnesol, building a calibration line with their standards.

The first observation is that both compounds were detected in all seed extracts obtained with the different solvents, as well as in the essential oil. The amount of β-caryophyllene and farnesol was 22.0 mg/g and 38.0 mg/g in EO, respectively, which corresponds to 2.2% and 3.8% *w*/*w* (Table 2). The quantity of β-caryophyllene was higher in EO than in the ethanol extract obtained following the traditional method. The quantitative decreasing order was: EO >> EthE > HexE > ChlE ≥ EtAE ≥ MetE. In part, the amount of farnesol was high in both essential oil and ethanol extracts, but different from β-caryophyllene, its amount was also considerable in all other extracts, such as ChlE, EtAE, and MetE (Table 2). Sequential solvent extractions resulted in a total yield of 4.5 mg/g caryophyllene and 20.4 mg/g farnesol. Essential oil and ethanol extract were found to be the richest preparations of terpenoids.

These data are consistent with previous studies on the essential oil of *Pterodon emarginatus*, which have shown the presence of an appreciable amount of the representative sesquiterpene β-caryophyllene [23,24,25,26,27]. Furthermore, other constituents have been detected in other studies, such as α-copaene [26], allo-aromadendrene [24,26], germacrene D and spathulenol [23], and farnesol [23,26]. In general, few studies are available in the literature on *Pterodon emarginatus* extracts and their phytoconstituents. Therefore, since this class of compounds has multiple biological activities, this research also focused on the actions of two selected compounds, β-caryophyllene and farnesol.

### 2.3. Total Phenolic and Flavonoid Levels in Pterodon emarginatus Extracts

Due to their beneficial effects on health, phenols are considered valuable components of medicinal plants. In particular, free radical scavenging activity is considered of interest in the prevention of several human illnesses. In this context, the detection of the levels of phenols and flavonoids in *Pterodon emarginatus* extracts was performed. Figure 3 shows that the methanol extract has the highest amount of both phenolic (A) and flavonoid (B) compounds compared to the ethanol or ethyl acetate extracts. The total phenolic content values of ethanol, ethyl acetate, and methanol extracts were 21.60 ± 4.74, 66.42 ± 8.09, and 267.70 ± 10.45 GAE mg/g extract (Figure 3A), respectively, while the total flavonoid content values reported in the same order were 9.88 ± 2.76, 11.29 ± 1.43, and 21.99 ± 0.75 QE mg/g extract (Figure 3B), respectively. Differently, the amount of phenols and flavonoids in essential oil and n-hexane and chloroform extracts could not be measured.

### 2.4. Radical Scavenging Activity of Pterodon emarginatus Extracts, Caryophyllene, and Farnesol

In light of the good presence of phenolic compounds, antiradical activity was determined using two types of assays that have different antiradical molecular mechanisms. In detail, antioxidant activity with the single electron transfer (SET) mechanism was estimated with the DPPH assay [28], while hydrogen atom donor ability was detected with an HAT-based method named the ORAC assay [28]. The results of the DPPH test are shown in Figure 4A. Although ascorbic acid, the standard antioxidant (positive control), has excellent scavenger capacity, it is important to note that most extracts can achieve the maximum amount of antiradical activity. The IC_50_ values were 111.3 µg/mL (MetE), 179.5 µg/mL (EO), 336.3 µg/mL (EtAE), 476.5 µg/mL (ChlE), and 629.4 µg/mL (EthE), while the values for the hexane extract and the compounds β-caryophyllene and farnesol were not obtained because they had very low activity (<15% inhibition). Thus, the order of potency obtained is: methanol extract > essential oil > ethyl acetate extract > chloroform extract > ethanol extract >>> hexane extract ≥ caryophyllene ≥ farnesol. These findings indicate that the free radical scavenging activity of the isolated constituents β-caryophyllene and farnesol is lower than that of the *Pterodon emarginatus* extracts, suggesting that the activity is due to other compounds or that there may be a synergistic effect between the extract components. The synergy between various types of extracts containing phenols and specific compounds has been explored by several authors, highlighting various degrees of cooperation in antioxidant activity [29,30,31]. The ORAC assay is used to measure the ability of antioxidants to break down the radical chain by monitoring the inhibition of peroxyl radical-induced oxidation. It is important to note that peroxyl radicals are the predominant radicals in lipid oxidation in human organisms. The ORAC test confirmed the significant scavenging activity of the methanol extract (12,624 ± 987 µmol TE/g) compared to other extracts such as EthE (8796 ± 1038 µmol TE/g), EtAE (7380 ± 568 µmol TE/g), HexE (6590 ± 785 µmol TE/g), EO (5790 ± 390 µmol TE/g), ChlE (5625 ± 675 µmol TE/g), or β-caryo-plyllene (1656 ± 527), and farnesol (1901 ± 152 µmol TE/g), in part in agreement with the results of the DPPH test (Figure 4B). Therefore, the strength of the scavenging activity detected by the ORAC assay is the following: methanol extract >> ethanol extract > ethyl acetate extract > hexane extract > EO ≥ chloroform extract >> farnesol ≥ caryophyllene.

These results can be explained by considering the synergistic effect of extracts containing multiple constituents compared to the activity of a single component [32]. Other authors reported synergism among the phenolics of cranberry with other bioactive substances such as ellagic acid and rosmarinic acid in redox modulation, increasing the antimutagenic effectiveness of the plant extract [29,33]. Another case is the synergism between phenolic compounds (i.e., rutin) and carotenoids (i.e., lutein or lycopene), which protects against LDL oxidation [34]. The possibility of synergistic antiradical activity due to caryophyllene and/or farnesol with other constituents of *Pterodon emarginatus* extracts may be of interest and explored in future investigations.

### 2.5. HT-29 Cell Viability

The MTT assay was used to assess the potential safety of compounds and extracts by investigating their effect on cell viability [35]. Four extracts *of Pterodon emarginatus*, caryo-phyllene, and farnesol were investigated in the concentration range of 0.01 µg/mL to 50 µg/mL to determine their effects on HT-29 cell proliferation. Figure 5 shows that the isolated compounds and extracts do not exhibit any cytotoxic activity up to 10 µg/mL. In fact, only at the higher concentration of 50 µg/mL, corresponding for β-caryophyllene and farnesol at about 250 µM, significant cytotoxic effects were observed. It is interesting to note that methanol and ethanol extracts have a lower impact on cell proliferation compared to other extracts or isolated compounds. Consistent with current data, other authors reported that β-caryophyllene and farnesol did not exhibit any cytotoxic effects up to 10 µg/mL when tested on African green monkey kidney cells (RC-37 cells) [36].

### 2.6. In Silico Absorption, Distribution, Metabolism, Excretion, and Toxicity (ADMET) Profile

By studying the pharmacokinetic parameters of pharmacological agents in silico, it is possible to accelerate and identify the best candidates for drug development [37]. Therefore, both compounds, β-caryophyllene and farnesol, were evaluated using the ADMETlab web tool [38,39,40]. Table 3 shows the ADMET profiles of both sesquiterpenes, which show elevated intestinal absorption and a high volume of distribution with moderate blood–brain barrier permeability. Both compounds, especially β-caryophyllene, are substrates of various cytochromes P450, making them susceptible to metabolism and interaction with other co-administrated drugs (Table 3). The short half-life could be challenging for long-term treatments, but appropriate drug delivery methods can overcome this characteristic. The toxicity profiles for both compounds can be considered adequate for human use, although farnesol has a warning about possible hepatotoxicity (Table 3).

β-Caryophyllene and farnesol are found in various aromatic plants used in food preparation and are therefore sometimes introduced in small amounts into the human organism [41,42]. Generally, they are considered safe, at least at the concentrations commonly found in the spices used as ingredients in foods [43]. molecules-28-07494-t003_Table 3Table 3In silico ADMET parameters of β-caryophyllene and farnesol.CompoundsMWg/molLogPBBBpPgpCYPsSubstrateHIAPPB%VDL/kgCLmL/min/kgT_1/2_Toxicity**β-Caryophyllene**cyclic sesquiterpene204.1905.906ModerateNon-substrateCYP1A2, CYP2C19, CYP2C9, and CYP2D6Elevate95.284.1389.94LowEye irritant**Farnesol**acyclic sesquiterpene222.2005.979LowNon-substrate,moderate inhibitorCYP1A2 inhibitorElevate89.065.6214.24LowH-HT, skin sensitizer, and eye irritantADMET: adsorption, distribution, metabolism, excretion, toxicity; MW: molecular weight; LogP: logarithmic of the octanol/water partition coefficient; BBBp: human blood–brain barrier permeability; Pgp: P-glycoprotein (MDR1 or 2 ABCB1); CYPs: cytochrome P450 (CYP) enzymes; HIA: human intestinal absorption (>30%); PPB: human plasma protein binding; VD: human volume of distribution; CL: clearance; T_1/2_: half-life (low < 3 h); H-HT: human hepatic toxicity. The pharmacokinetic parameters were obtained using the ADMETlab platform [44].

## 3. Materials and Methods

### 3.1. Chemical Reagents

Aluminium chloride, 2,2′-azobis(2-amidinopropane) dihydrochloride (AAPH), Folin-Ciocalteu phenol reagent, fluorescein, gallic acid, 3-(4,5-dimethylthiazol-2-yl)-2,5-diphenyl-2H-tetrazolium bromide, and 1,3,5-triisopropylbenzene were purchased from Merck KGaA (Darmstadt, Germany). Acetic acid, dimethylsulfoxide, methanol, n-hexane, chloroform, ethyl acetate, and phosphate-buffered saline were acquired from Merck (Milan, Italy). Water purified was obtained with a MilliQ water purification system (Millipore, Burlington, MA, USA). The purity of the reference standards was ≥98%.

### 3.2. Pterodon emarginatus: Plant Material

The seeds from mature fruits were collected from May to August 2020, near the city of Goiás, Bahia State, Brazil. The samples were confirmed by Dr. Fabricio Mendes Miranda (Southwest Bahia State University, UESB, Brazil) according to the deposited voucher specimens (PE 0015). A reference drug sample (PE0015 A) is deposited at DPPS, University of Padova (Italy). The air-dried seeds were 1.8 to 2.2 cm in length, with an average weight of 0.707 ± 0.023.

The EO of *Pterodon emarginatus* was obtained by steam distillation with a Clevenger apparatus (Buchi distillation unit K-314, Cornaredo, Italy) from chopped air-dried seeds (30 g) for 2 h in triplicate, with a yield of 1.5% *w*/*w*. To facilitate the separation of components from the water, diethyl ether was used, which was then removed in a stream of nitrogen. Finally, the EO was stored at a temperature of 4 °C, away from light, in a glass container. An ethanol preparation was prepared as described in traditional Brazilian medicine [6]. In detail, it was obtained from chopped air-dried seeds (10 g) by performing an extraction with 96% ethanol (50 mL) for 24 h; the solvent was removed by a stream of nitrogen, obtaining a yield of 40.9% *w*/*w*. The ethanol extract (EthE) was then stored at a temperature of 4 °C, away from light, in a glass container. Furthermore, sequential extraction was chosen to assess the polarity of the compounds present within the air-dried seeds of *Pterodon emarginatus*, to assess the possibility of separating certain categories of compounds from the total constituents, and also to study the activity in relation to that of the total extract used in traditional medicine (EthE). Thus, four solvents with different polarities (n-hexane, chloroform, ethyl acetate, and methanol) were used sequentially to extract the seeds (10 g). The extraction was carried out for 24 h with each solvent (50 mL), and the extracts obtained were dried in nitrogen current. The yields (*w*/*w*) were 16.9% in hexane, 16.7% in chloroform, 1.4% in ethyl acetate, and 2.0% in methanol, with a residual plant matrix of 63.0%. The extracts were then stored at a temperature of 4 °C, away from light, in glass containers.

### 3.3. GC Chromatography Analysis

The chromatographic analysis of the extracts was performed with a gas chromatograph (Thermo Trace DSQ, Waltham, MA, USA) equipped with a DB5 column (Thermo, Waltham, MA, USA) coupled with a quadrupole mass detector. For the GC qualitative analysis, the oven was programmed from 70 to 120 °C at 5 °C/min rate, held at 120 °C for 8 min, followed by a 2 °C/min ramp from 120 to 150 °C, held for 1 min at 150 °C, followed by a third ramp from 150 to 200 °C at 10 °C/min, and finally, held for 1 min. The inlet temperature was kept at 250 °C. The electron ionization source temperature was set at 240 °C. Identification was performed by comparing their mass spectra with those from the NIST Mass Spectral Database [45]. The match and reverse match exceeded 890 in all samples for all detected compounds. Methanol was used to dissolve all extracts at a concentration of 5 mg/mL, while essential oil was solubilized at 5 mg/L. A precise aliquot of 1,3,5-triisopropylbenzene, a volatile compound that is certainly not present in the natural extracts, is added to the samples to be injected as an internal standard (IS) [46,47]. β-Caryophyllene and farnesol were quantified by calibration curves, which were experimentally constructed with correlation coefficients close to unity.

### 3.4. TPC and TFC Assays

Total phenolic content (TPC) was determined using the Folin-Ciocalteu phenol reagent [48]. The reaction mixture containing the sample solution, Folin-Ciocalteu reagent, and sodium carbonate (22% *v*/*v*) was kept in the dark at room temperature for 2 h. The absorbance was measured at 760 nm (Beckman Coulter model DU 800, Fullerton, CA, USA). Gallic acid (GA) was used to obtain the standard curve. TPC was expressed as milligrams of GA equivalents per g of extract (GAE mg/g). Total flavonoid content (TFC) was detected using the aluminum chloride colorimetric method [49]. The sample solution was added to aluminum chloride (25% *w*/*v*) and incubated for 15 min at room temperature. The absorbance was detected at 425 nm with a Beckman Coulter DU 800 instrument (Fullerton, CA, USA). Quercetin was used to obtain the standard curve. Data were expressed as mg of quercetin equivalents per g of substance (QE mg/g).

### 3.5. Free Radical Scavenging Assays

The DPPH scavenging assay was performed according to the previously reported method [50]. Aliquots of DPPH (70 µM) were subdivided into vials, and then sample solutions (0.1 µg/mL–10 mg/mL) were added. Subsequently, the vials were incubated for 60 min in the dark at 25 °C. Finally, the samples were detected spectrophotometrically at 517 nm (Beckman Coulter DU 800, Fullerton, CA, USA). The oxygen radical absorbance capacity (ORAC) assay allows for evaluating the ability of substances to interfere with oxidative reactions induced by peroxidic radicals [51]. Briefly, aliquots of 0.08 µM fluorescein were mixed with each sample solution (10–50 µg/mL) or PBS (blank) or 6-hydroxy-2,5,7,8-tetramethylchroman-2-carboxylic acid solution (6.25–50 µM trolox). The samples were then incubated at 37 °C for 10 min. Successively, the oxidative reaction was started with 0.15 M 2,2′-azobis(2-amidinopropane)-dihydrochloride (AAPH), and the decrease in fluorescence was recorded for 60 min at 37 °C (Victor Nivo Multimode microplate reader, Waltham, MA, USA). The ORAC values were expressed as TEAC (Trolox Equivalent Antioxidant Capacity, µmol TE/g of substance).

### 3.6. Measurement of Cell Viability

Cell viability was assessed with an MTT assay [52]. Human Caucasian colon adenocarcinoma (HT-29) cells were grown in RPMI 1640 medium containing 10% fetal bovine serum (Merck, Darmstadt, Germany) and maintained in sterile flasks, placed in an incubator at 37 °C (5% CO_2_ atmosphere). Cells were seeded in 96-well plates at a density of 5000 and allowed to grow for 24 h. Subsequently, cells were treated with each compound or extract or with medium (control). After 24 h of incubation, cells were washed with RPMI 1640 medium to avoid any interference with the assay and subsequently treated for 4 h with a 0.05 mg/mL 3-(4,5-dimethylthiazol-2-yl)-2,5-diphenyl-2H-tetrazolium bromide solution [36]. During incubation, viable cells react with MTT to produce purple formazan crystals, which can be solubilized with 2-propanol. Chloroform and ethyl acetate extracts were not tested due to problems of solubility in the cell incubation medium and the low quantity obtained during the extraction process; this was mainly for EtAE. The absorbance was measured using a Victor Nivo multimode microplate reader (Waltham, MA, USA) at a wavelength of 570 nm.

### 3.7. Prediction of ADMET Properties

In silico estimated characteristics of absorption, distribution, metabolism, excretion, and toxicity (ADMET) of the two sesquiterpenes were achieved with the pharmacokinetic web tool ADMETlab [44]. The parameters considered were the logarithmic ratio of the partition coefficient (LogP), the ability to cross the human blood–brain barrier (BBBp), the interactions with the P-glycoprotein (MDR1 or 2 ABCB1) and cytochrome P450 enzymes, the passage through human intestinal absorption, the human oral bioavailability, the binding to plasma human proteins, the human volume of distribution, the clearance, and the half-life of the compounds considered. Data on the prediction of human toxicity are also reported in Table 2.

### 3.8. Statistical Analysis

Each value is reported as the mean ± SEM of 3–8 independent experiments. The results were analyzed with Microsoft Excel for Windows 10, while sigmoid curve fitting and statistical evaluations were performed using GraphPad Prism 8 (San Diego, CA, USA). The half-maximum inhibitory concentration (IC_50_) was calculated by nonlinear regression [50]. The statistical significance between the control and each treatment was evaluated using the Student’s *t*-test, while comparisons among three or more groups were performed with ANOVA, followed by Tukey’s multiple comparison test. The level of significance was established at *p* < 0.05.

## 4. Conclusions

The extracts of *Pterodon emarginatus* contain various volatile components, including α-copaene, β-caryophyllene, germacrene, spathulenol, and farnesol. Thus, sesquiterpenes constitute the most characteristic portion of the volatile component found in essential oil and in ethanol and methanol extracts, as well as in n-hexane, chloroform, and ethyl acetate extracts. Among *Pterodon emarginatus* terpenes, β-caryophyllene and farnesol were quantified in all preparations and evaluated in vitro with respect to their free radical scavenging activity and cytotoxicity. The results show that the methanol extract exhibited the highest antiradical activity; indeed, all extracts have appreciable antioxidant activity detected with DPPH and ORAC assays. The MTT test revealed that *Pterodon emarginatus* extracts have no significant cytotoxic activity except at the highest concentration of 50 µg/mL, which is similar to that observed for β-caryophyllene and farnesol. This observation is in agreement with the in silico study, which does not predict relevant tissue toxicity. Furthermore, in silico investigation shows that β-caryophyllene and farnesol are well absorbed through the intestinal tract but may have a high first-pass effect that can reduce their bioavailability. Passing through the blood–brain barrier is also predictable. Importantly, the half-life is short, indicating a short persistence in the human body after a single administration, which could be a limitation in chronic treatments. The Italian Health Ministry’s list of herbal substances that are allowed or not in food supplements does not include this Brazilian plant; therefore, *Pterodon emarginatus* has the potential to be a new botanical remedy. In general, the set of data obtained in this investigation supports the interest in *Pterodon emarginatus* seed extracts and their traditional use, mainly as ethanol preparations. The antiradical activity can be explained by the substantial presence of phenolic compounds, such as flavonoids and sesquiterpenes, which are higher in methanol and ethanol extracts. The relevance of a more thorough clinical evaluation is suggested by the antioxidant activity of the different extracts, together with their very low cytotoxicity.

## Figures and Tables

**Figure 1 molecules-28-07494-f001:**
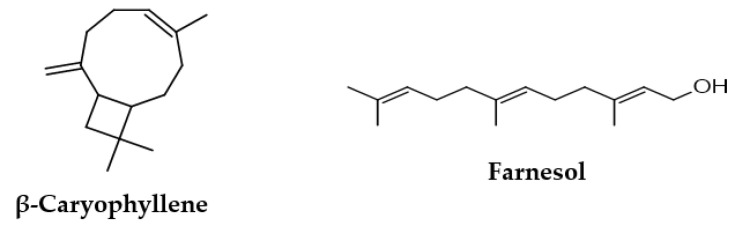
Chemical structures of β-caryophyllene and farnesol, two characteristic compounds detected in essential oil and extracts obtained from *Pterodon emarginatus* seeds.

**Figure 2 molecules-28-07494-f002:**
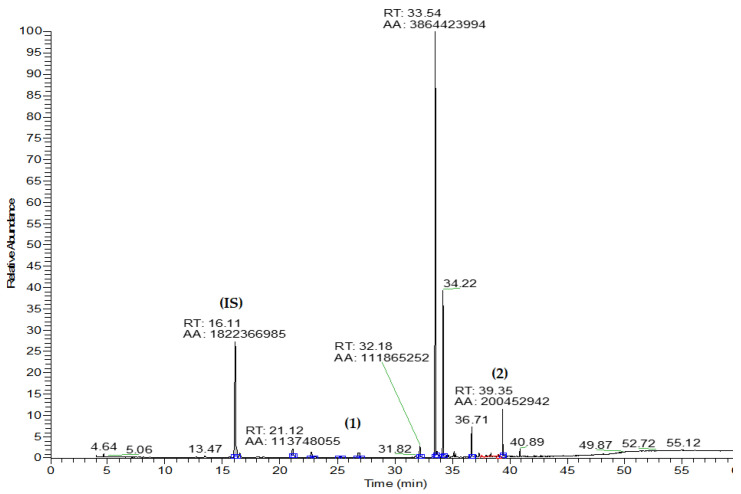
An example of a gas chromatogram of essential oil obtained from the seeds of *Pterodon emarginatus*. (IS): internal standard (1,3,5-triisopropylbenzene, Rt = 16.11); (1): β-caryophyllene (Rt = 26.88 min); (2) farnesol (Rt = 39.35 min).

**Figure 3 molecules-28-07494-f003:**
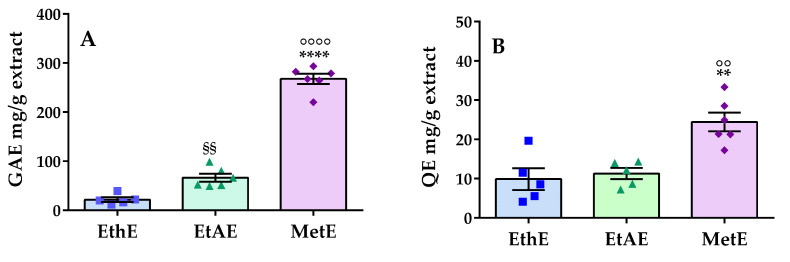
The phenolic (**A**) and flavonoid (**B**) contents of various *Pterodon emarginatus* extracts, including the ethanol extract (EthE), obtained with the method used in traditional Brazilian medicine, and the ethyl acetate and methanol extracts (EtAE and MetE), obtained with sequential extraction. For essential oil (EO), n-hexane, and chloroform extracts, the quantity of phenols and flavonoids was not measurable. The data are the mean ± SEM of 5–6 experiments. GAE: gallic acid equivalents; QE: quercetin equivalents; ^§§^: *p* < 0.01 versus EthE (**A**); **: *p* < 0.01 versus EthE (**B**); °°: *p* < 0.01 versus EtAE; °°°°: *p* < 0.0001 versus EthE; ****: *p* < 0.0001 versus EtAE.

**Figure 4 molecules-28-07494-f004:**
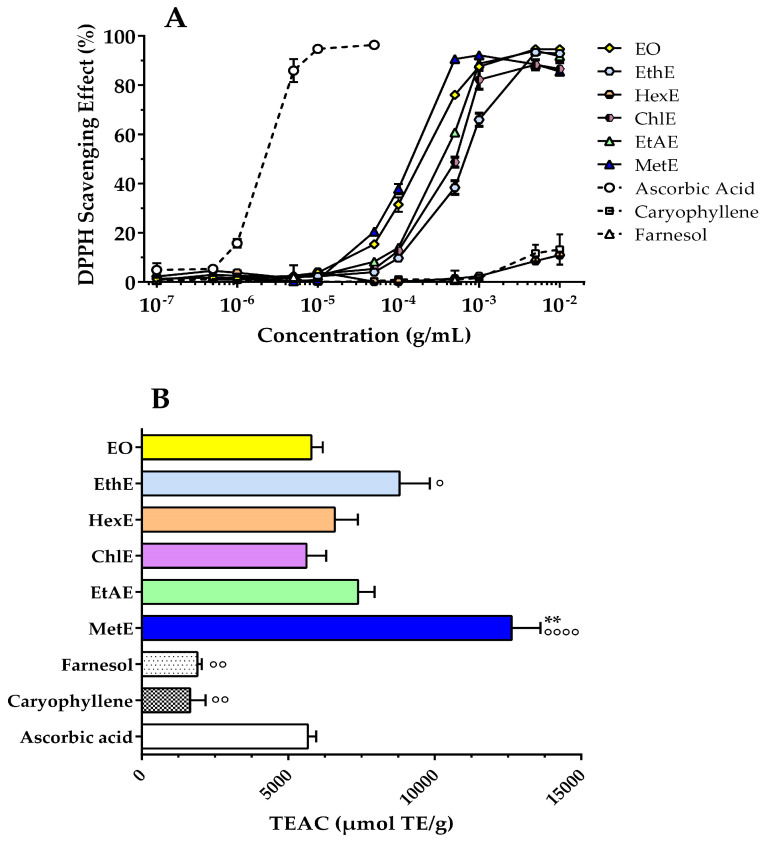
Radical scavenging activity of essential oil (EO), ethanol extract (EthE), hexane extract (HexE), chloroform extract (ChlE), ethyl acetate extract (EtAE), and methanol extract (MetE) of *Pterodon emarginatus* seeds, and of β-caryophyllene and farnesol detected by DPPH (**A**) and ORAC (**B**) assays. Ascorbic acid was the positive control for antiradical activity. Undisclosed SEM fall within the respective symbols (**A**). The ORAC values are expressed as TEAC (Trolox Equivalent Antioxidant Capacity), μmol of Trolox equivalents per gram of each extract or compound. **: *p* < 0.01 methanol extract versus each extract or compound; °°: *p* < 0.01 β-caryophyllene or farnesol versus ascorbic acid and versus each extract; °: *p* < 0.05 EthE versus ascorbic acid; °°°°: *p* < 0.0001 MetE versus ascorbic acid.

**Figure 5 molecules-28-07494-f005:**
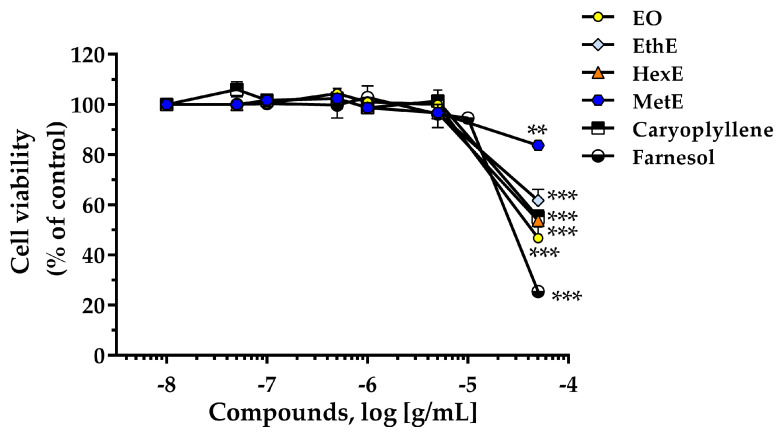
The effects of essential oil (EO), ethanol extract (EthE), hexane extract (HexE), and methanol extract (MetE) of *Pterodon emarginatus* seeds, and caryophyllene and farnesol on HT-29 cell viability were detected by the MTT assay. **: *p* < 0.01, and ***: *p* < 0.001 versus control (not treated cells). Chloroform and ethyl acetate extracts were not tested due to problems with their solubility in cell medium and the low quantity obtained during the extraction process (EtAE).

**Table 1 molecules-28-07494-t001:** Chemical composition of the extracts of *Pterodon emarginatus* seeds.

Compounds	R_T_	EO	EthE	HexE	ChlE	EtAE	MetE
		% *					
α-Copaene	21.12	6.24	14.90	4.55	1.53	1.25	1.45
β-Caryophyllene	26.88	6.92	12.6	3.37	1.01	0.53	0.64
Allo-Aromadendrene	32.18	6.10	13.14	3.73	--	--	--
Germacrene D	33.54	212.10	350.85	109.50	25.87	--	1.75
Germacrene B	34.22	69.73	115.02	37.59	7.85	--	--
Spathulenol	36.71	8.07	28.81	6.26	6.91	8.32	3.94
Farnesol	39.35	11.00	110.89	16.69	23.70	23.30	16.46
Farnesyl acetate	45.57	--	156.27	15.69	25.45	--	--

R_T_: retention time; *: percentage of each phytoconstituent obtained by comparison with the internal standard (1,3,5-triisopropylbenzene); EO: essential oil; EthE: ethanol extract; HexE: n-hexane extract; ChlE: chloroform extract; EtAE: ethyl acetate extract; MetE: methanol extract; --: not detectable.

**Table 2 molecules-28-07494-t002:** Quantitative detection of β-caryophyllene and farnesol in *Pterodon emarginatus* seed extracts.

Compounds	EO p/p		EthE p/p		HexE p/p		ChlE p/p		EtAE p/p		MetE p/p	
	mg/g	%	mg/g	%	mg/g	%	mg/g	%	mg/g	%	mg/g	%
β-Caryophyllene	22	2.2	2.8	0.28	1.8	0.2	<1.0	<0.1	<1.0	<0.1	<1.0	<0.1
Farnesol	38	3.8	20	2.0	4.4	0.4	5.6	0.6	5.8	0.6	4.6	0.6

## Data Availability

Data are contained within the article and Appendix A.

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
