# Peer review of "Pterodon emarginatus* Seed Preparations: Antiradical Activity, Chemical Characterization, and In Silico ADMET Parameters of β-caryophyllene and Farnesol"

_molecules, 2023, doi:10.3390/molecules28227494_

Round 1
Reviewer 1 Report
Comments and Suggestions for Authors
The manuscript by Froldi et al. is devoted to biologically active compounds from the seeds of the Brazilian plant Pterodon emarginatus extracted with different solvents or obtained as an essential oil after steam distillation. The strong point of the work is detailed characterization of bioactive properties of the obtained extracts and in silico prediction of the toxicity of their two main constituents. Generally, this study makes a good impression, properly organized, and well written. It is of interest for the readers and can be recommended for publication after clarification of some issues related mainly to the methodological aspects:
1. The plant material should be described more thoroughly. An information on the seed size, degree of maturity, moisture and ash content, and sample weight and averaging procedure should be provided.
2. The authors claim that major compounds were quantified by GC-MS. However, there is no information in the Experimental section on the analytical standards used in the quantification procedure, as well as the nature and role of the internal standard mentioned in the manuscript.
3. Identification of the compounds presented in Table 1 must be confirmed by the corresponding mass spectra (in supplementary material) and their comparison with those found in MS library (at least similarity index should be provided in Table 1). Moreover, for all the detected compounds except farnesol and caryophyllene the term tentative identification should be used since reliable identification requires analytical standards or other methods to confirm the structure of the identified analytes.
4. Since sequential extraction with different solvents (n-hexane, chloroform, ethyl acetate, and methanol) was used, it would be useful to provide total amounts of the extracted compounds for these four solvents in Table 2.
Minor comments:
Line 340. Please, use proper term “electron ionization” instead of outdated “electronic impact ionization”.
Figure 5. The X-axis name should be concentration, not compound.
Comments on the Quality of English Language
Only minor editing of English language required.
Author Response
The authors thank the reviewer for very useful comments that helped improve the manuscript. As suggested, the manuscript has been amended by adding more details on the characteristics of Pterodon emarginatus seeds, the extraction methods and gas chromatography analysis. The Word revision tool has been used to highlight all changes.
- To provide the details required, the information on Pterodon emarginatus seeds has been included in the Results and Methods.
- The analytical standard used in the quantification procedure (1,3,5-triisopropylbenzene) and its role as an internal standard (IS) have been described in the Materials and Methods (GC chromatography analysis) and reported in the Results sections of the manuscript. Its mass spectrum was added to the Supplementary materials (Figure S8).
- The identification of the detected compounds presented in Table 1 has been confirmed by the corresponding mass spectra compared with those found in the MS library, as reported in Materials and Methods (GC chromatography analysis section). In all samples, the match and reverse match exceeded 890 for all the compounds that were detected. Supplementary materials have been updated with the mass spectra of the compounds detected (Figures S6-S12).
- The suggestion has been greatly appreciated. The total amounts of β-caryophyllene and farnesol obtained by extraction with the four solvents used in succession have been added to the Results and Discussion section of the manuscript.
Minor comments:
- Line 340. The text was revised in accordance with the suggestion.
- Figure 5. The x-axis is labelled “Compounds” because it refers to the different compounds and extracts studied, while the concentrations are expressed in logarithms. In general, the square bracket is used to specify “concentration”. We are confident that the reviewer will agree with the explanation.
The authors thank the reviewer for the valuable suggestions that helped improve the manuscript.
Reviewer 2 Report
Comments and Suggestions for Authors
Guglielmina Froldi et al reported that “Pterodon emarginatus seed preparations: Antiradical Activity, Chemical Characterization, and in Silico ADMET Parameters of β-caryophyllene and farnesol”. Since it is questionable whether β -caryophyllene and farnesol are involved in the antiradical activity, it would be better to publish the paper after a major revision.
Major comments
Line 206
"These findings indicate that the free radical scavenging activity of the isolated constituents β- caryophyllene and farnesol is lower than that of the extracts, suggesting a synergistic effect among the numerous components of the plant-derived mixture."
There are cases of synergistic effects, but “synergistic effect” is an overexpress based on the measurement of only two components. Is there really no possibility that there is a small amount of the active ingredient?
It is also possible that β-caryophyllene and farnesol are not involved at all and there is a synergistic effect between other compounds. It is known that the main components that react in DPPH and ORAC are phenols.If β-caryophyllene and farnesol, for example, are to have a synergistic effect with these compounds, experimental data showing higher DPPH and ORAC values when β-caryophyllene and farnesol are added compared to phenol compounds alone would be necessary.
Author Response
The authors thank the reviewer for comments and suggestions that helped to improve the manuscript. The Word revision tool has been used to highlight all changes.
The point is certainly important. The authors agree that this research has not shown a synergistic effect between caryophyllene (or farnesol) and another constituent present in the extracts. A possible synergistic effect between two or even more compounds present in the extracts remains an intriguing starting point for further research. The manuscript has been modified to address this point, highlighting the need for future research to study a possible synergy between caryophyllene (or farnesol) and other compounds present in extracts of Pterodon emarginatus. Thank you for your comment.
Round 2
Reviewer 1 Report
Comments and Suggestions for Authors
The authors clarified the issues and revised the manuscript accordingly. Now I can recommend it for publication in its present form.
Comments on the Quality of English LanguageEnglish is fine, only minor editing is required in some cases.
Reviewer 2 Report
Comments and Suggestions for Authors
The previously noted points have been changed. I recommend this paper for acceptance.
My personal opinion is that β-Caryophyllene and Farnesol have low activity in ORAC and DPPH and are not effective in synergizing with other compounds to increase their activity. But this is just my opinion without experimental evidence, I thought it would be good to put two things (other compounds or synergistic effect) as possibilities.